# Twisted moiré photonic crystal enabled optical vortex generation through bound states in the continuum

Tiancheng Zhang [1,9], Kaichen Dong [2,3,4,5,6,9] ✉, Jiachen Li[1,3], Fanhao Meng [1,3], Jingang Li [7], Sai Munagavalasa[8], Costas P. Grigoropoulos [7], Junqiao Wu [1,2,3] & Jie Yao [1,2,3] ✉

The twisted stacking of two layered crystals has led to the emerging moiré physics as well as intriguing chiral phenomena such as chiral phonon and photon generation. In this work, we identified and theoretically formulated a non-trivial twist-enabled coupling mechanism in twisted bilayer photonic crystal (TBPC), which connects the bound state in the continuum (BIC) mode to the free space through the twist-enabled channel. Moreover, the radiation from TBPC hosts an optical vortex in the far field with both odd and even topological orders. We quantitatively analyzed the twist-enabled coupling between the BIC mode and other non-local modes in the photonic crystals, giving rise to radiation carrying orbital angular momentum. The optical vortex generation is robust against geometric disturbance, making TBPC a promising platform for well-defined vortex generation. As a result, TBPCs not only provide a new approach to manipulating the angular momentum of photons, but may also enable novel applications in integrated optical information processing and optical tweezers. Our work broadens the field of moiré photonics and paves the way toward the novel application of moiré physics.

Moiré patterns formed in twisted bilayer two-dimensional van der Waals materials have led to the observation and exploration of emerging electronic properties[1–5], including moiré excitons[6], fractional Chern insulators[7], and Mott insulators[8,9]. Along with the surge of research into moiré bilayers of two-dimensional materials, moiré patterns in photonics have also drawn attention from the optical community[10–14], with demonstrations of twist-induced flat bands[15], energy localization[16], and optical soliton[17]. Moreover, the twist of photonic structures breaks the mirror symmetry and enables chiral optical properties. Here we show that the structural chirality of twisted

bilayer photonic crystal (TBPC), through the bound states in the continuum (BICs) in both layers of photonic crystals (PhCs), enables the chirality of light: the optical vortex.

An optical vortex is a light beam with a spiral phase front and an undefined phase at the beam center, resulting in a "doughnut" intensity profile. Optical vortices have been shown to carry orbital angular momentum (OAM), which is a new degree of freedom beyond the spins of light. Since the discovery of light's OAM[18], optical vortices have greatly expanded the horizon of numerous research fields[19–24], including super-resolution imaging[25,26], optical communication[27–29],

[1]Applied Science and Technology Graduate Group, University of California, Berkeley, CA 94720, USA. [2]Department of Materials Science and Engineering, University of California, Berkeley, CA 94720, USA. [3]Materials Sciences Division, Lawrence Berkeley National Laboratory, Berkeley, CA 94720, USA. [4]Tsinghua-Berkeley Shenzhen Institute, Tsinghua Shenzhen International Graduate School, Tsinghua University, Shenzhen 518055, China. [5]Institute of Data and Information, Tsinghua Shenzhen International Graduate School, Tsinghua University, Shenzhen 518055, China. [6]Center of Double Helix, Tsinghua Shenzhen International Graduate School, Tsinghua University, Shenzhen 518055, China. [7]Department of Mechanical Engineering, University of California, Berkeley, CA 94720, USA. [8]Department of Chemical and Biomolecular Engineering, University of California, Berkeley, CA 94720, USA. [9]These authors contributed equally: Tiancheng Zhang, Kaichen Dong. ✉e-mail: dkc22@sz.tsinghua.edu.cn; yaojie@berkeley.edu

micromanipulation[30,31], and quantum information processing[32,33]. As a result, optical vortex generation has attracted extensive attention.

In this work, we discover that the twisted stacking of two-layer PhCs enables the generation of robust optical vortex radiation that is insensitive to either the incident angle or the illumination position of the incoming light. By taking advantage of the special interlayer coupling mechanism between the two twisted PhC slabs, a twist-enabled channel for energy transfer between free space and the bound states in the continuum (BICs)[34–37] in the monolayer PhC is created. Upon arbitrary Gaussian beam illumination onto the TBPC, at-$\Gamma$ BIC mode in PhC 1 is excited by its coupled guided resonances in PhC 2 via interlayer channeling. The BIC mode in turn excites the radiative guided resonances in PhC 2 for optical vortex emission by inducing Pancharatnam-Berry (PB) phases, as we will show in more detail below. We theoretically and numerically demonstrate the "doughnut" intensity distribution and vortex phase distribution in the generated beam from the TBPC system with a high Q factor, which makes TBPC a promising platform for vortex micro-laser applications. Moreover, optical vortex radiation with both even and odd order and at any wavelength can be realized by adjusting the TBPC design.

## Results
### Methodology
Figure 1a shows a schematic diagram of the system considered in this work. The device is constructed by stacking two planar PhC slabs with a small twist. Each slab is a honeycomb array of silicon nanodisks. Below we show that upon excitation by an arbitrary pulsed Gaussian beam, the at-$\Gamma$ BIC modes (herein referred to as BIC for simplicity) in both PhC slabs will be excited, which further radiate energy to the +z and −z directions. Such counterintuitive excitation of BIC modes by Gaussian beam and radiation from BIC modes are realized by a unique interlayer channel created by the moiré structure. Specifically speaking, in a monolayer PhC slab with twofold rotational symmetry, radiative resonant modes can couple to the far-field with frequencies and in-plane wave vectors matching the plane waves in free space. Those radiative resonant modes are usually called guided resonances[38].

Meanwhile, any singlet at $\Gamma$ point is an exception since the rotational symmetry of any plane wave with in-plane wave vector $(k_x, k_y) = 0$ mismatches the rotational symmetry of that at $\Gamma$ singlet[34]. Thus, such a singlet is isolated from the free space, and we call this resonant mode which is decoupled from the free space a symmetry-protected BIC. Even with another identical PhC slab introduced to form a bilayer PhC system, the at-$\Gamma$ mode will remain a BIC mode, due to the protected two-fold symmetry.

In bilayer PhC systems, when the two PhC layers are perfectly aligned with zero twist angle, the periodicity remains the same as that of monolayers. Thus, the at-$\Gamma$ BIC modes in both layers are still isolated from guided resonances due to the phase mismatch mechanism. However, the introduction of a small twist between the two layers generates a moiré structure[1], which breaks the short-range periodicity of a monolayer PhC. Since the rigorous phase match mechanism rely on the short-range periodicity, it will be broken by introducing moiré structure to the PhCs. The BIC modes in one slab can thus couple to the guided resonances in the other slab which is coupled to the free space[15], as shown in the following paragraphs. This interlayer coupling mechanism enabled by the twists in TBPC systems allows energy transfer between BIC modes and the free space via guided resonances, so it is named as a "moiré channel" in this work (Fig. 1b). Moreover, the states of polarization (SOPs) of far-field radiation from these guided resonances are momentum-space variant, and they will form a polarization vortex around the BIC in the momentum space[34–36]. Hence, when BIC modes radiate energy via these guided resonances enabled by the moiré channel, vortices emerge in the real-space radiation.

### Moiré channel in TBPC
Here we prove the existence of the moiré channel, i.e., with non-zero twist angles, the coupling between the BIC in one layer and the guided resonances in the other layer is finite. Coupled-mode theory is used as it has been proven to be accurate in analyzing the photonic behaviors of a TBPC system[15]. Our theory begins with a well-defined transverse electric (TE) mode in the silicon disk[15]. When two disks are placed closely, the crosstalk between the eigenmodes in either disk is

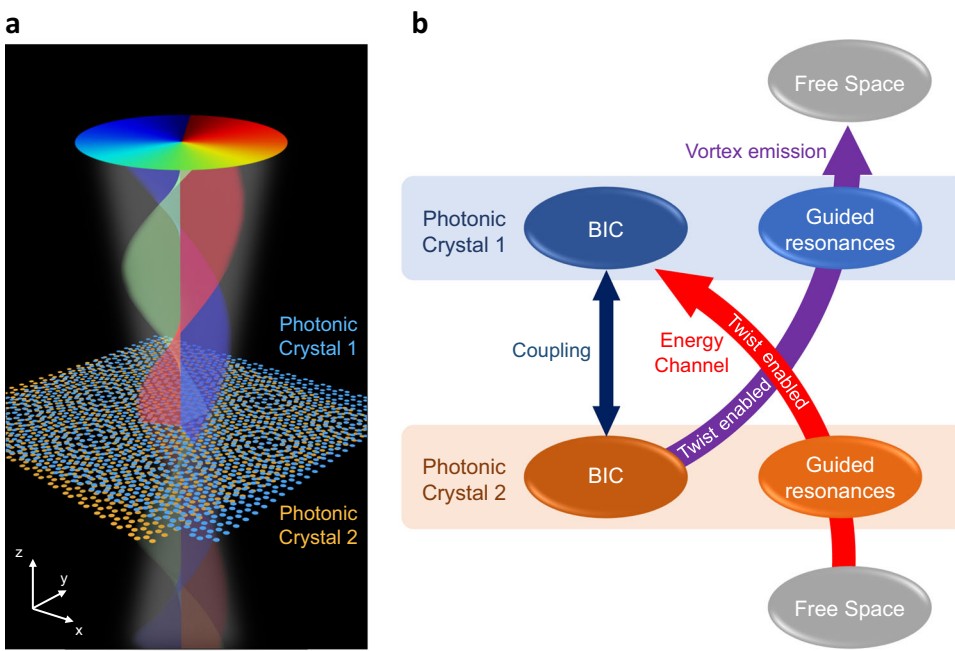

**Fig. 1 | The twisted bilayer photonic crystal (TBPC) system. a** Schematic of TBPC with optical vortex emitted from the AA stacking regions to both +z and −z directions. The height and diameter of the disks are 220 nm and 450 nm, respectively. The lattice constant of monolayer honeycomb PhC is 1.0 μm. **b** Schematic view of the twist-enabled energy transfer between free space light beams and at-$\Gamma$ BIC modes through guided resonances.

described by:

$$\begin{cases} \frac{da_1(t)}{dt} = (i\omega_1 - \kappa_1)a_1 + ig_{12}a_2 \\ \frac{da_2(t)}{dt} = (i\omega_2 - \kappa_2)a_2 + ig_{21}a_1 \end{cases} \qquad (1)$$

where $a$, $\omega$, and $\kappa$ are the mode amplitude, the angular eigenfrequency, and the decay rate, respectively. Note that $\omega_1 = \omega_2 = \omega_0$, $\kappa_1 = \kappa_2 = \kappa_0$ for identical disks. $g_{12}$ and $g_{21}$ are the coupling coefficient between the two modes. Without loss of generality, we set $g_{12} = g_{21} = g$[15].

For a monolayer PhC consisting of such a disk array, the traveling mode with the wave vector $\mathbf{k}$ (mode $\mathbf{k}$ in short) can be denoted as a linear combination of each disk modes, and the amplitude $a_k(t)$ is: $a_k = \frac{1}{\sqrt{N}} \sum \exp(i\mathbf{k} \cdot \mathbf{r}_j)a_j$, where $N$ is the total number of the disks, and $\mathbf{r}_j$ is the real-space position of the center of disk $j$. We set the coupling intensity between BIC in PhC 2 and mode $\mathbf{k}_1$ in PhC 1 is $\zeta(\mathbf{k}_1)$. Specifically speaking, mode $\mathbf{k}_1$ is a BIC mode when $\mathbf{k}_1 = \mathbf{0}$. We can derive that for non-twist bilayer PhCs, inside the first Brillouin zone of the monolayer, $\zeta(\mathbf{k}_1)$ is always zero except $\mathbf{k}_1 = \mathbf{0}$ ($\Gamma$ point), which means BIC modes in PhC 2 only couple to BIC modes in PhC 1, as we can see in Fig. 2a.

However, in a TBPC system, $\zeta(\mathbf{k}_1)$ can be denoted as:

$$\zeta(\mathbf{k}_1) = \frac{i}{N} \sum_j \exp\left(i\mathbf{k}_1 \cdot \mathbf{r}_{1j}\right) \cdot g\left(\mathbf{r}_{1j}\right) \qquad (2)$$

where $N$ is the total number of the disks in one layer, $\mathbf{r}_{1j}$ is the real-space position of the center of disk $j$ in PhC 1, and $g\left(\mathbf{r}_{1j}\right)$ is the coupling intensity between disk $j$ in PhC 1 and its nearest neighbor in PhC 2. Since the moiré structure differs the coupling intensity $g(\mathbf{r}_{1j})$ for different $j$, the coupling strength $\zeta(\mathbf{k}_1)$ is non-zero at non-$\Gamma$ points. Hence, BIC modes in PhC 1 can couple to the guided resonances in PhC 2, and vice versa, as shown in Fig. 2b. Moreover, we prove that the coupling coefficient $\zeta(\mathbf{k}_1)$ is nearly identical for same $|\mathbf{k}_1|$ points (Supplementary Fig. 2), which means that the BIC in one layer can excite the guided resonance modes around BIC in the other layer with roughly the same phase and intensity, as illustrated in Fig. 2b.

**Optical vortex generation in TBPC**

In the following paragraph, we will analytically prove the optical vortex generation, which is radiated from the guided resonances excited by the BIC mode. Since honeycomb lattice is protected by the $C_2^Z T$

symmetry, the SOPs in the far-field will be linearly polarized[34], oriented in the direction $\theta(\mathbf{k}_1)$ to the $x$ axis. To get the explicit expressions of the radiated field, we start from the temporal coupled-mode theory (TCMT) main equations:

$$\begin{cases} \frac{dA(\mathbf{k}_1)}{dt} = [i\omega(\mathbf{k}_1) - \kappa(\mathbf{k}_1)]A(\mathbf{k}_1) + \zeta(\mathbf{k}_1)A_{BIC} \cdot \exp(i\omega_{BIC}t) \\ |E_{out}\rangle = \mathbf{D}A \end{cases} \qquad (3)$$

where $A(\mathbf{k}_1)$, $\omega(\mathbf{k}_1)$, and $\kappa(\mathbf{k}_1)$ are the amplitude, the eigenfrequency, and the decay rate (due to radiative loss) of mode $\mathbf{k}_1$ in PhC 1, respectively. $A_{BIC}$ and $\omega_{BIC}$ are the amplitude and the resonant frequency of the BIC mode in PhC 2, respectively. $\zeta(\mathbf{k}_1)$ is the coupling intensity between mode $\mathbf{k}_1$ in PhC 1 and the BIC mode in PhC 2 calculated in Eq. (2). $|E_{out}\rangle$ is the outgoing wave amplitude, and $\mathbf{D}$ is the coupling coefficient between the resonance and the outgoing plane wave. The orientation of the SOP should be[24]:

$$\begin{pmatrix} d_x \\ d_y \end{pmatrix} = \sqrt{d_x^2 + d_y^2} \cdot \begin{pmatrix} \cos(\theta(\mathbf{k}_1)) \\ \sin(\theta(\mathbf{k}_1)) \end{pmatrix} \qquad (4)$$

where $\theta(\mathbf{k}_1)$ is the angle of the polarization vector. By dividing this Jones vector in the basis of left- and right-handed circular polarized (LCP and RCP) unit vectors, the radiated field $|E_{out}\rangle$ of mode $\mathbf{k}_1$ can be formulated as:

$$|E_{out}\rangle = K\zeta(\mathbf{k}_1)e^{-i\theta(\mathbf{k}_1)}|L\rangle + K\zeta(\mathbf{k}_1)e^{-i\theta(\mathbf{k}_1)}|R\rangle \qquad (5)$$

where $K$ is an integrated parameter, which is nearly a constant if we choose a circular loop around the $\Gamma$ point[24]. $|L\rangle$, $|R\rangle$ denote the LCP and RCP unit vectors, respectively. From Fig. 2b, we notice that $\zeta(\mathbf{k}_1)$ is also roughly a constant in the circular loop around the $\Gamma$ point. Thus, $K\zeta(\mathbf{k}_1)$ shall not be able to introduce another phase factor. From Eq. (5) we notice that the emitted light will be composed of two non-trivial parts: LCP and RCP. Both parts of the light will gain a PB phase depending on the orientation of the SOP of the guided resonance: $\theta_{LCP} = -\theta(\mathbf{k}_1)$ and $\theta_{RCP} = \theta(\mathbf{k}_1)$, where $\theta_{LCP}$ and $\theta_{RCP}$ denote the phase of the LCP and RCP portion, respectively, and $\theta(\mathbf{k}_1)$ is the angle of the polarization vector. Since BIC is the crossing of nodal lines for $x$ direction and $y$ direction portions of the far-field radiation, the SOPs will present a vortex around the BIC[34–36]. Thus, the angle of polarization

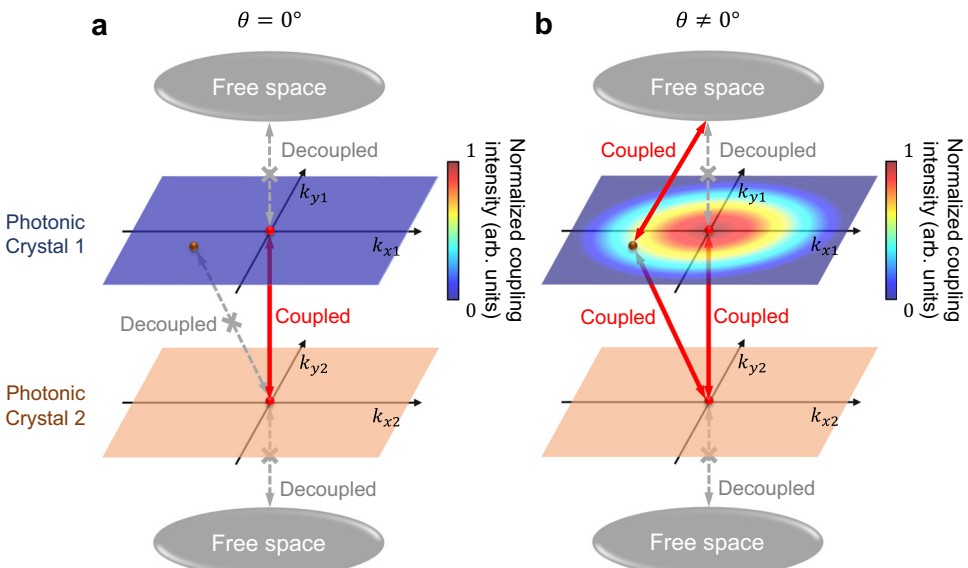

**Fig. 2 | Interlayer coupling features in bilayer PhCs for (a) perfectly aligned PhCs and (b) twisted bilayer photonic crystals (TBPCs).** The color of the upper plane denotes the absolute value of coupling intensity between the BIC in PhC 2 and the guided resonances with wave vector $(k_{x1}, k_{y1})$ in PhC 1. Note that there is no coupling between the BIC in PhC 2 and guided resonances in PhC 1 in **a**.

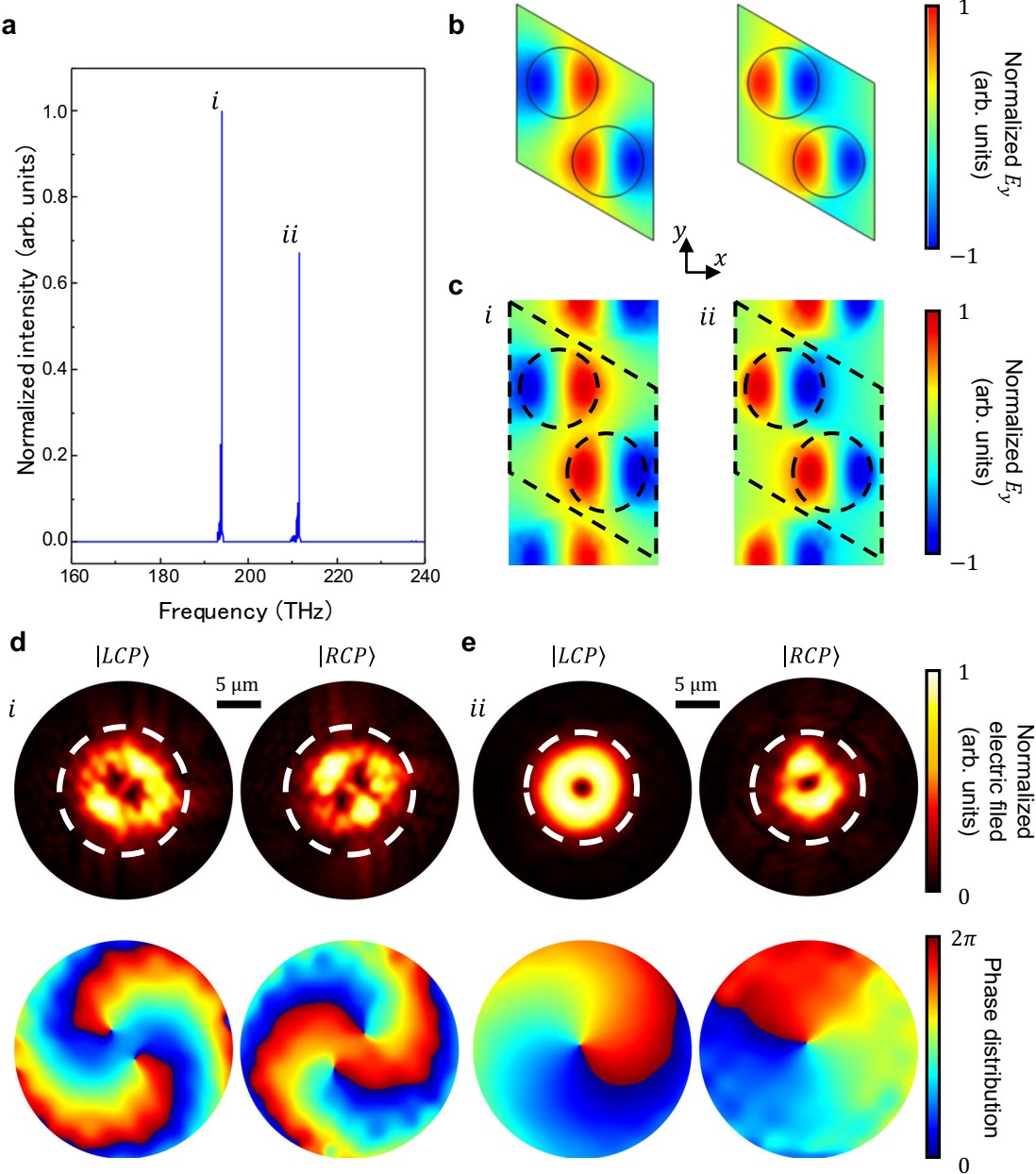

**Fig. 3 | Optical vortex generation in a representative twisted bilayer photonic crystal (TBPC) system. a** The mode intensity spectrum of the TBPC system. The two peaks in the spectrum are marked as "*i*" and "*ii*", with the frequency of 193.4 THz and 211.2 THz, and the Q-factor of $3.1 \times 10^3$ and $3.4 \times 10^3$, respectively. **b** Electric-field profile of the BIC modes hosted in monolayer PhC. **c** Electric-field profiles in PhC 1 of the TBPC system at peak "*i*" and "*ii*". **d**, **e** The electric-field distribution of the optical vortex emission 12 μm away from the TBPC surface at peak "*i*" (**d**) and "*ii*" (**e**). The upper panels depict the normalized amplitude of the LCP and RCP portions of the emission, while the lower panels are the phase distribution of the corresponding upper panels in dashed white circles.

vector $\theta(\mathbf{k}_1)$, as well as $\theta_{LCP}$ and $\theta_{RCP}$, will form a vortex structure around the $\mathbf{k}_1 = \mathbf{0}$ point. Hence, the emitted beam will gain the desired spiral phase front, whose topological order is the same as that of the BIC mode and can be described as $l = \pm q$. Here $q$ is the topological order of the BIC, minus and plus signs correspond to LCP and RCP portions, respectively.

To verify the above theory, we perform a full-wave numerical simulation of a representative TBPC with a 300 nm separation and a 1.5° twist angle. We excite the system with a pulsed Gaussian beam and measure the field profile on a plane centered at the AA point and is parallel to the TBPC surface after the pulse is over. The AA point denotes the center of AA stacking region where the top and bottom honeycomb PhC layers are well aligned. Simulation details can be found in the Method section. Two peaks are identified in the mode intensity spectrum (Fig. 3a), which are marked as "*i*" and "*ii*". The mode profile in the central cross-section in PhC 1 is shown in Fig. 3c, which matches the BIC mode profile simulated using a monolayer PhC (Fig. 3b). Then we examine the radiated beams corresponding to the two peaks ("*i*" and "*ii*") in the far field. As illustrated by Fig. 3d, e, both the "doughnut" shape in the amplitude profile and the clear vortex in the phase distribution diagram serve as clear evidence of OAM existence. The topological order $l$ equals to −2 and +1 for the RCP portions of peak "*i*" and peak "*ii*", respectively, which is consistent with the theoretical results of the SOPs around the BIC: the lowest topological

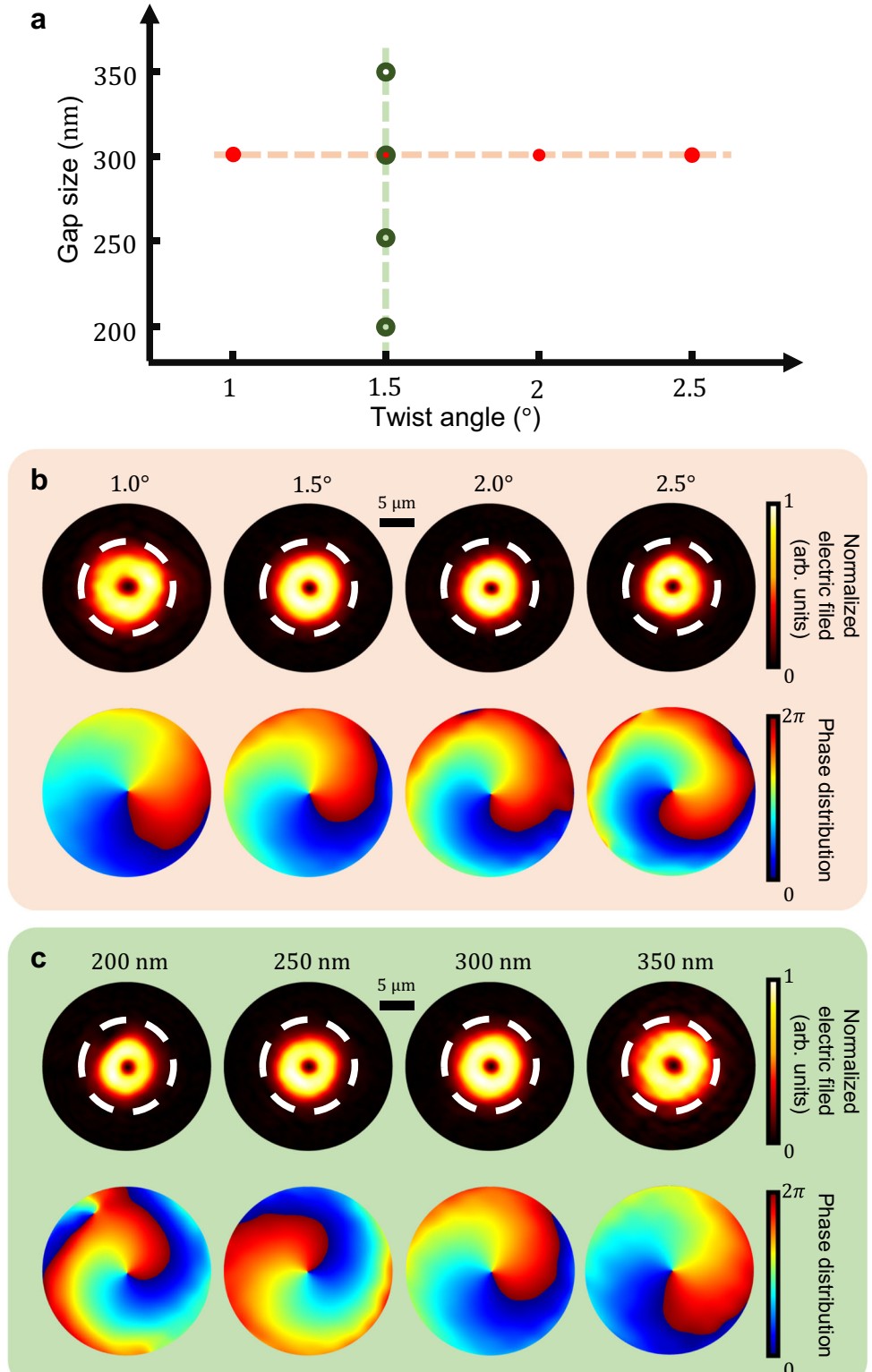

**Fig. 4 | Optical vortex generation as functions of various interlayer separation and twist angles. a** The parameter space of interlayer separation and twist angle. The interlayer separation ranges from 200 nm to 350 nm, and the twist angle ranges from 1.0° to 2.5°. The red dots and green circles denote the parameters we choose for the results in **b**, **c**, and the dashed lines are guides to the eye.
**b** Normalized amplitude and phase distribution at the red dots in **a**. **c** Normalized amplitude and phase distribution at the green circles in **a**.

order of the at-Γ BIC in a honeycomb lattice could be either −2 or 1[24]. Hence, all the full-wave numerical simulation results not only justify our theoretical model, but also directly proves the emission of the optical vortex from the AA region and further demonstrate the important role of BIC mode in the OAM generation. Compared with previous micro-optical vortex generators[24], the optical vortex generated by TBPC is not limited to even numbers.

Furthermore, we change the interlayer separation and the twist angle between the two layers. Since the interlayer coupling between two PhC slabs does not change the SOPs around BIC in the monolayer, the topological orders of the emitted vortices will remain the same. We perform a numerical calculation based on the simulated system for Fig. 3 but change its interlayer separation and twist angle following Fig. 4a. For tidiness, only the LCP portion profiles at peak "ii" are included, but all the following conclusions also work for the RCP portion and the peak "i". As shown in Fig. 4b, c, the topological orders of the vortices are 1 in all cases, proving that the optical vortex emission is robust against disturbance. Therefore, the TBPC system is insensitive to imperfections in experiments, such as fabrication error and thermal expansion effects, making it a stable platform for realizing optical vortices.

Identification of the temporal sequence of the energy transfer in the aforementioned modes provides straightforward evidence of the optical vortex generation mechanism. We demonstrate the time evolution procedure of optical vortex generation where the TBPC system is excited by a pulsed Gaussian beam with a random incident position. At time=0 ps, the Gaussian beam just reaches the TBPC system (Fig. 5a). The Gaussian pulse excites all possible modes, including some low-Q modes in the TBPC system, leading to a chaotic amplitude profile in the monitored plane (Fig. 5c). However, since the low-Q guided resonances dissipate much faster than the BIC mode, the radiation by the BIC mode will gradually appear against the chaotic background as all unwanted low-Q modes vanish after 5 ps. It will form a beautiful optical vortex emission with a zero-intensity point as well as the phase singularity. We notice that the center of the emission is at the AA point rather than the center of the incident beam. It demonstrates that the vortex emission is insensitive to either the incident angle or the illumination position of the incoming light. Consequently, no real-space alignment of the position and incident angle is needed in real applications. This is because the optical vortex emission is excited by the self-contained BIC modes rather than the incident light. More direct demonstration of the robustness of optical vortex generation in TBPC can be found in Supplementary Fig. 6. Thus, in contrast to previous work[24], our system has much fewer requirements for the incident light source, making TBPC a promising robust platform for stable vortex generation. Moreover, it is possible to incorporate the TBPC system with micro/nanoelectromechanical systems (MEMS/NEMS) for adjustment of both interlayer separation and twist angle. This could lead to the creation of tunable vortex beams with adjustable quality factors, vortex center positions, divergence angles, etc. Furthermore, with the TBPC system, a tunable OAM laser with adjustable order numbers could be possible.

In summary, we theoretically show the existence of optical vortex emission in a bilayer photonic crystal system with non-zero twist angles. An interlayer channeling model is formulated based on coupled-mode theory to demonstrate that the optical vortex emission originates from the twist-enabled coupling between the bound state in the continuum (BIC) mode and the guided resonances. Moreover, the optical vortex generation in TBPC is robust against disturbance of geometric parameters, making TBPC a promising platform for well-defined vortex generation, and providing strategies to design stable vortex lasers. Besides the application in vortex generation, the twist-enabled coupling mechanism in TBPC provides a tunable interlayer channel to connect BIC modes to the free space. This will not only benefit the development of BIC study, but also broaden the field of moiré photonics. Due to the intrinsic similarities between photonic systems and condensed matter systems, this work may also guide the exploration of OAM in moiré van der Waals structures.

## Methods
### Theoretical analysis
Please see the Supplementary Information for the derivations.

### Simulations
The electric-field distribution in Fig. 3b, c are calculated using COMSOL Multiphysics from COMSOL Inc. Bloch boundary conditions are applied in the x and y boundaries, while perfect matched layers (PMLs) are applied in the z direction. Without loss of generality, the refractive index of the silicon disks is set to be 3.47. The electric-field amplitude and phase distribution of optical vortices are calculated using commercial finite-difference time-domain software (Lumerical FDTD solutions from Lumerical Inc.). We place the TBPC system perpendicular to the vertical axis (z axis) and place the AA point at the center of the simulation area. In order to match the experiments, we use a pulsed Gaussian beam as the excitation and apply PMLs in all directions. The electric-field distribution of emission is probed by a frequency-domain field profile which is 12 μm away from the TBPC.

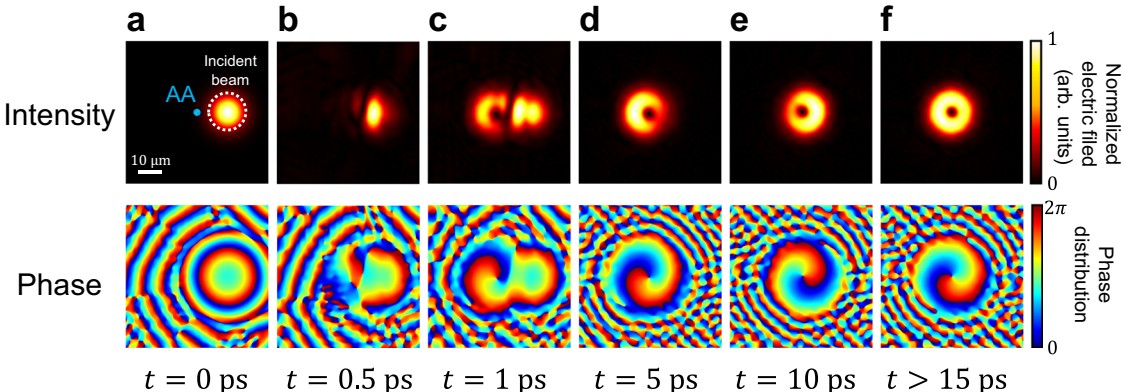

**Fig. 5 | Time evolution of the real-space profile. a–f** Depict normalized amplitude and phase distribution of the LCP portion of the emission at different times after excitation. The blue dot in **a** denotes the position of the AA point, and the white dashed circle denotes the position of the incident Gaussian beam.

## Data availability

The data that supports the plots within this paper are available on Zenodo (https://doi.org/10.5281/zenodo.8201070). All other data used in this study are available from the corresponding authors upon request.

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

## Acknowledgements

We thank D. Cui and Y. Kang for the discussions. The work was supported by Heising–Simons Faculty Fellowship. The simulation part was supported by the U.S. NSF Grant no. DMR2140304. Work in C.G.'s LTL lab was supported by the National Science Foundation through Grant CMMI-2024391. K.D. acknowledges the start-up funding in Tsinghua Shenzhen International Graduate School (SIGS), Tsinghua University.

## Author contributions

J.Y. conceived the basic idea for this work. T.Z. and J.Y. gave a theoretical explanation. K.D., T.Z. and Jingang L. designed the structures and performed the numerical simulations. K.D., T.Z., Jiachen L., F.M. and S.M. analyzed the simulated data. J.Y. supervised the project. T.Z. and K.D. wrote the draft of the manuscript. All authors discussed the results and contributed to the manuscript.

## Competing interests

The authors declare no competing interests.
