## [Peer Review File · Nature Communications]

REVIEWER COMMENTS

Reviewer #1 (Remarks to the Author):

Zhang et al. propose an original mechanism to generate an optical vortex in a photonic lattice with the use of twisted moiré photonic crystal. Specifically, the authors show that for non-zero twisted angles, the symmetry-protected bound state in the continuum (BIC) of a single-layer photonic crystal would be enabled to radiate to the free space in the form of an optical vortex beam via inter-layer coupling with guided mode resonances (GMRs). This mechanism is explained within the framework of coupled mode theory and is numerically demonstrated by full-wave simulations. This work highlights a new functionality of moiré photonics and is timely with the rise of this field. I really appreciate the authors' effort to explain the microscopic mechanism underlying the creation of these chiral vortex beams. The work can be considered for publication in Nature Communications after the authors address the following issues/comments:

1. I am a bit disappointed that there are no quantitative results derived from the coupled mode theory model. The parameters (κ_1 , κ_2 , g_{12} , g_{21} ...) can be extracted from the simulations of a single particle (Fig.1 in the Supplemental Information) and bi-atomic particles when two particles are placed at a distance $a=1\mu\text{m}$. The coupling strength between disks belonging to separate layers can be extracted using the same technique. Then, with all parameters of the tight-binding well-defined, the authors can provide some quantitative comparisons between the moiré modes obtained by the analytical models and those given by the simulation results (COMSOL).
2. Does the mode emitting an optical vortex in this work correspond to the moiré band that can be rendered flat in the authors' previous work (10.1103/PhysRevLett.126.223601)?
3. Which parameters of the system define the vortex size (approximately given by dashed white circles in Fig.3d and Fig.4b, c)?
4. The authors claim that their optical vortices are "topologically protected." What kind of topology are they referring to? While the vortex may be robust against geometrical parameters and incident angles, I do not see any mechanism that can preserve the vortex creation if some defects or disorders are introduced into the system.
5. If the twist angle becomes zero, the BIC nature of the mode will be recovered. Can we use the term quasi-BIC here? I expect that $Q \sim 1/\theta^2$, with θ being the twisted angle; is this true?

6. Recent progress in the field of moiré photonics with bilayer photonic crystal should be included; three references (with one self-citation) for such emerging topics are too few. For flatband generation with moiré bilayers, the authors should include recent works on twisted 2D photonic moiré (10.1038/s41377-021-00601-x, 10.1038/s41377-022-00977-4) and period mismatched 1D photonic moiré (10.1103/PhysRevResearch.4.L032031). The authors should explain the differences between this work and the previous report on the chirality of moiré bands at the Gamma point (10.1103/PhysRevLett.126.136101) and the recent report on quasi-BIC with moiré photonic crystal (10.1103/PhysRevLett.128.253901). The authors also need to explain their novelty compared to the chiral quasi-BICs proposed for bilayer photonic crystals (10.1103/PhysRevLett.126.073001).

Reviewer #2 (Remarks to the Author):

This is a very interesting study. It involves coupling between two honeycomb lattices, where the guided resonance of one honeycomb layer excites the free-space mode, while the BIC mode (with wavevector $K=0$) of the other honeycomb layer is excited by the guide resonance from the former layer. As the $k=0$ mode has an intrinsic topological charge, the radiated free space mode results in optical vortices. This study is the first to discover optical vortices radiating from optical moiré lattices. Rigorous simulation confirmed this finding. I support its publication.

I believe, however, that this work could be presented in a more transparent manner. Please see some of the points mentioned below.

Regarding my understanding, the formation of vortices in the far field is rooted in the BIC at $k=0$, as the radiated vortices have the same topological charge as the BIC mode. If this is indeed the case, I suggest that it be explicitly written.

I was initially confused about the difference between BIC and guided resonance. However, I think now I understand that BIC belongs to the guided resonance mode, except that the BIC has a zero wavevector.

With respect to Figure 3, the charge 2 vortices seem already split into two well-separated single-charged vortices? Is this due to the instability of the higher-order vortex at the very beginning (BIC at $K=0$), or this is due to the twisting between layers? Besides, since the twisting angle does not change the birth of the vortices in the far-field, then what happens when the twisting angle takes a much larger one (at the present plots, only small angles are considered).

Reviewer #3 (Remarks to the Author):

This paper presents a theoretical study of a photonic crystal bilayer, twisted in a manner that mimics twisted bilayer graphene (TBLG). In recent years, TBLG has been of substantial interest in condensed matter physics, because its tunable flat electronic bands can give rise to a wide range of behaviors including superconductivity. The vast majority of TBLG's interesting properties do not carry over into photonics. Superconductivity, for instance, requires an inter-particle pairing mechanism that has no natural analogue in photonics. What the authors propose, here, is that twisted photonic crystals can generate optical vortex beams. This happens because the the twist induces a pair of bound-state-in-the continuum (BIC) slab modes to couple to free space modes in a particular fashion.

As far as I know, the radiative mechanism detailed in this paper, involving what the authors call the "Moiré channel", is novel. It is characterized using coupled-mode theory and verified using full-wave simulations. The theoretical analysis appears to be sound, though too many details are relegated to the supplemental materials, which makes the discussion in the main text difficult to follow. For example, in the discussion around Eqs. (3)-(4), it would be better to provide additional schematics clarifying the definition and role of the SOPs and $\theta(k)$, and to fill in some intermediate steps.

There is a notable related prior work. The generation of an optical vortex from a photonic crystal slab was reported in an experimental paper published in Nature Photonics in 2020 [19], by Bo Wang et al. That team also exploited some of the same properties of BICs in photonic crystal slabs, but the precise mechanism was different and not involve Moiré patterns.

Given the existence of at least one other way to generate an optical vortex from a slab geometry, it should be necessary to demonstrate or argue for a clear advantage for the present method. The authors do not appear to make much efforts in this. They do note that the present method can generate vortices of odd order, whereas Ref. [19] is limited to even orders, but given the brevity, it is hard to evaluate how significant this difference is.

The authors also note that the twist is "tunable", but I find this point dubious. In the condensed-matter context, the twist geometry has attracted attention as it provides a relatively simple way to generate new materials (i.e., by making another sample with a different twist angle), bypassing the limitations of chemistry. In photonics, however, there is already substantial freedom in designing and fabricating photonic structures, and numerous ways to exert active control over these structures. Therefore, it is

unclear that varying the twist angle is superior to other methods, such as having a slab with 3D patterning.

If anything, bilayer configurations might be a harder to fabricate, at least in the nanophotonic regime. The design studied by the authors is certainly not one that can be directly implemented in an experiment, since it consists of suspended silicon disks. Alongside the absence of any experimental results, the practicality of the idea is a significant concern.

In summary, this is a paper with credible results, but one would expect more work to be done to convince the reader that the design provides a performance advantage over previous designs --- including Ref. [19], but also other possible vortex emitters. Such discussions should be presented in sufficient detail before the paper can be considered for publication in Nature Communications.

Reviewer #1 (Remarks to the Author):

Zhang et al. propose an original mechanism to generate an optical vortex in a photonic lattice with the use of twisted moiré photonic crystal. Specifically, the authors show that for non-zero twisted angles, the symmetry-protected bound state in the continuum (BIC) of a single-layer photonic crystal would be enabled to radiate to the free space in the form of an optical vortex beam via inter-layer coupling with guided mode resonances (GMRs). This mechanism is explained within the framework of coupled mode theory and is numerically demonstrated by full-wave simulations. This work highlights a new functionality of moiré photonics and is timely with the rise of this field. I really appreciate the authors' effort to explain the microscopic mechanism underlying the creation of these chiral vortex beams. The work can be considered for publication in Nature Communications after the authors address the following issues/comments:

Response: We thank the reviewer for the time and effort in reviewing our paper. We are also grateful that the reviewer appreciates the novelty of this work. We have carefully worked on all the helpful comments and suggestions. The point-by-point responses with corresponding revisions are as follows:

1. I am a bit disappointed that there are no quantitative results derived from the coupled mode theory model. The parameters (κ_1 , κ_2 , g_{12} , g_{21} ...) can be extracted from the simulations of a single particle (Fig.1 in the Supplemental Information) and bi-atomic particles when two particles are placed at a distance $a=1\mu\text{m}$. The coupling strength between disks belonging to separate layers can be extracted using the same technique. Then, with all parameters of the tight-binding well-defined, the authors can provide some quantitative comparisons between the moiré modes obtained by the analytical models and those given by the simulation results (COMSOL).

Response: We thank the reviewer for this important comment! Quantitative comparison between analytical results and simulation results are important for verifying the theoretical model. The correspondence between the coupled mode theory model and simulation results in twisted bilayer photonic crystal (TBPC) systems has been quantitatively proven in our previous paper [R1].

In order to make a comparison between the analytical and numerical results in this work, we obtained the divergence angle of the generated vortex beam in the far field as a function of the twist angle using both approaches. As a key feature of vortex beams, the divergence angle is critical for the future vast applications of the generated optical vortex [R2-R4]. Furthermore, different from the topological order or the eigenfrequency, which are mainly defined by the single-layer properties, the divergence

angle is directly connected to the interlayer coupling mechanism. Comparing the analytically and numerically calculated divergence angles based on extracted parameters will further confirm the reliability of our theoretical model.

First, the analytical calculation is conducted as follows: According to the coupled mode theory, the BIC in PhC 2 will couple to the guided resonances \mathbf{k}_1 in PhC 1, and the corresponding coupling intensity $\zeta(\mathbf{k}_1)$ is:

$$\zeta(\mathbf{k}_1) = \frac{i}{S_c} \iint \exp(i\mathbf{k}_1 \cdot \mathbf{r}_1) \cdot g(\mathbf{r}_1) d^2r_1 \quad (\text{R1})$$

where $g(\mathbf{r}_1)$ is the coupling intensity of the interlayer nearest neighbor (NN) disks at \mathbf{r}_1 point. Based on the simulation of a single particle and single-layer photonic crystal, we can extract that: $\kappa_1 = \kappa_2 = 9.12$ THz, and $g_{12} = g_{21} = 3.35$ THz. Similarly, the function $g(\mathbf{r}_1)$ can also be calculated, based on which the interlayer coupling strength $\zeta(\mathbf{k}_1)$ can be plotted, as is shown in Fig. S2 of the Supplementary Information.

Here, the profile of the function $\zeta(\mathbf{k}_1)$ is also plotted in Fig. R1 (left column), showing that the range of $\zeta(\mathbf{k}_1)$ in the reciprocal space increases when the twist angle goes up. We note that the guided resonances \mathbf{k}_1 will further couple to the far field with frequencies and in-plane wave vectors matching the plane waves in free space. As a result, when the range of $\zeta(\mathbf{k}_1)$ increases, the range of in-plane wave vectors of the plane waves in free space also increases, leading to a larger divergence angle of the generated optical vortex beam. Therefore, we can deduce the divergence angle analytically by evaluating the $\zeta(\mathbf{k}_1)$ function. We need to note that, at $\mathbf{k}_1 = 0$ point (Γ point), it is a BIC mode, which does not couple to free space modes. Therefore, although $\zeta(\mathbf{k}_1)$ is non-zero at Γ point, the radiation intensity is zero, and thus the radiation intensity distribution is in the form of a doughnut shape.

Secondly, by simulating the xz-plane beam profile in Lumerical FDTD, we can numerically get the divergence angle, as shown in Fig. R1 (right column).

Fig. R1 | Analytically calculated normalized interlayer coupling intensity (left column) and numerically simulated xz -plane beam profile (right column) as functions of twist angles. The twist angles are 1.0° (a), 1.5° (b), 2.0° (c), 2.5° (d), and 3.0° (e), respectively. The twisted bilayer photonic crystal (TBPC) is centered at $z = 0$ plane in the right column.

With the above results, we compare the analytically calculated and simulated divergence angles in Fig. R2. The numerically calculated divergence angle has the same trend as the analytical results: the divergence angle increases at larger twist angles. The slight discrepancy could be further decreased by considering higher orders of

coupling in the theoretical model as well as using finer meshes in the numerical simulation.

Fig. R2 | Comparison between analytically calculated and numerically simulated divergence angles.

In response to this comment, all the above results and discussion have been added to the Supplementary Information, Section F, where the moiré modes obtained by the analytical models and those given by the simulation results are thoroughly and quantitatively compared.

Reference

- [R1] Dong, Kaichen, et al. "Flat bands in magic-angle bilayer photonic crystals at small twists." *Physical review letters* 126.22 (2021): 223601. <https://doi.org/10.1103/PhysRevLett.126.223601>
- [R2] Chung, Hyeongju, et al. "Generation of E-band metasurface-based vortex beam with reduced divergence angle." *Scientific Reports* 10.1 (2020): 1-8. <https://doi.org/10.1038/s41598-020-65230-7>
- [R3] Vallone, Giuseppe, et al. "General theorem on the divergence of vortex beams." *Physical Review A* 94.2 (2016): 023802. <https://doi.org/PhysRevA.94.023802>
- [R4] Ma, Hongyu, et al. "Broadband vortex beams generation with narrow divergence angle using polarization insensitive metasurface." *IEEE Access* 8 (2020): 218062-218068. <https://doi.org/10.1109/ACCESS.2020.3042236>

2.Does the mode emitting an optical vortex in this work correspond to the moiré band that can be rendered flat in the authors' previous work (10.1103/PhysRevLett.126.223601)?

Response: We appreciate this thoughtful comment by the reviewer. Though both this work and our previous PRL paper (10.1103/PhysRevLett.126.223601) discuss TBPC system with two similar photonic structures, the modes we analyze in these two papers are completely different:

- (1) The mode in this work is the hybridization of at- Γ BIC modes and guided resonances close to the Γ point of the Brillouin zone.
- (2) The mode in our previous PRL paper is the hybridization of guided resonances close to the K and K' points of the Brillouin zone.

Hence, the two modes analyzed in these two works are different with distinct physical properties.

3.Which parameters of the system define the vortex size (approximately given by dashed white circles in Fig.3d and Fig.4b, c)?

Response: This is a nice point. According to our simulation results, the key parameter that defines the vortex size is the twist angle.

In Fig. 4 of the main text, we measure the generated optical vortex at the xy-plane that is 12 μm away from the TBPC surface. Also, in the above Fig. R1, we show the simulation result at the xz-plane to illustrate the influence of the twist angle on the vortex size as a function of the distance from the TBPC surface. With results in those two figures, it can be concluded that: when the twist angle is small, the vortex beam has a relatively large waist width and a small divergence angle; as the twist angle increases, the waist width of the vortex beam goes down with a larger divergence angle. If we only compare the vortex sizes at a distance of 12 μm away from the TBPC surface (Fig. 4), the vortex size will decrease with larger twist angles.

4.The authors claim that their optical vortices are "topologically protected." What kind of topology are they referring to? While the vortex may be robust against geometrical parameters and incident angles, I do not see any mechanism that can preserve the vortex creation if some defects or disorders are introduced into the system.

Response: Thanks for pointing out this confusion. This term refers to the topological order of the vortex beam that is generated by the TBPC system.

Since the optical vortex generation in our system is rooted in the BIC modes, the order of the generated vortex is defined by the topological order of those BIC modes. Hence,

if the BIC modes remain the same, the order of the generated vortex will be protected, regardless of defects or disorders in TBPC.

The above definition of “topologically protected” is different from conventional meaning of this term. In order to avoid possible confusion, we have deleted the phrase “topologically protected” in our manuscript.

5.If the twist angle becomes zero, the BIC nature of the mode will be recovered. Can we use the term quasi-BIC here? I expect that $Q \sim 1/\theta^2$, with θ being the twisted angle; is this true?

Response: We appreciate these insightful comments by the reviewer.

(1) Yes, the term quasi-BIC may be used here. However, for clarification, we did not use this term in our manuscript because of the following reasons:

First, as shown in Fig. 2, with zero or non-zero twist angles, the bilayer photonic crystal system will have totally different properties: With zero twist angle, though the BIC mode exists, there is no moiré channel between two photonic crystal layers and thus no optical vortex emission is observed. On the other hand, with non-zero twist angle, the optical vortex is generated by TBPC thanks to the moiré channel. Therefore, our discovery of optical vortex generation in TBPC is directly related to the moiré channel rather than the single-layer BIC mode. If the term “quasi-BIC” is used, readers would have the wrong impression that the mode for vortex generation is similar to a BIC mode at zero twist angle. We hope to emphasize the dramatic change in the moiré channel and optical vortex generation when the twist angle changes from zero to non-zero. Therefore, in order to avoid possible confusion, we did not use the definition of “quasi-BIC”.

Secondly, in this work, the BIC mode specifically refers to the BIC mode in a single layer. Adding the term “quasi-BIC” might distract the readers and they may get confused when discussing single-layer BIC and twisted bilayer quasi-BIC simultaneously. As a result, we did not use the term quasi-BIC in our paper.

(2) Yes, we also expect that the Q factor is proportional to $1/\theta^2$. As we can see in Fig. R1, the area of the interlayer coupling strength $\zeta(\mathbf{k}_1)$ is proportional to θ^2 , which means that the number of moiré channels created by the twist structure is approximately proportional to θ^2 . Assuming the total decay rate is proportional to the number of moiré channels, the Q factor, which is inversely proportional to the decay rate, should satisfy $Q \sim 1/\theta^2$.

However, the above conclusion requires an infinitely large TBPC system, while in real experiments (and also numerical simulations) the lateral size of the TBPC system is limited. In such a finite TBPC system, the energy coming from localized optical modes

will be partially emitted through the lateral edges of TBPC in xy-plane, as shown in Fig. R3, leading to a dramatical decrease in the experimental/numerical Q factor. To experimentally observe the phenomena of $Q \sim 1/\theta^2$, the TBPC system should be large enough with many superlattices [R5].

Fig. R3 | Schematic of a TBPC system showing the lateral energy emission. The background picture is the xy-plane projection of the TBPC system. The blue circles denote the silicon disks, and the AA point of the moiré superlattice is located at the center. The orange box denotes the lateral edges of this finite TBPC system, where partial energy coming from the localized mode is emitted (white arrows).

Reference

[R5] Grepstad, Jon Olav, et al. "Finite-size limitations on quality factor of guided resonance modes in 2D photonic crystals." *Optics express* 21.20 (2013): 23640-23654. <https://doi.org/10.1364/OE.21.023640>

6. Recent progress in the field of moiré photonics with bilayer photonic crystal should be included; three references (with one self-citation) for such emerging topics are too few. For flatband generation with moiré bilayers, the authors should include recent works on twisted 2D photonic moiré (10.1038/s41377-021-00601-x, 10.1038/s41377-022-00977-4) and period mismatched 1D photonic moiré (10.1103/PhysRevResearch.4.L032031). The authors should explain the differences between this work and the previous report on the chirality of moiré bands at the Gamma point (10.1103/PhysRevLett.126.136101) and the recent report on quasi-BIC with moiré photonic crystal (10.1103/PhysRevLett.128.253901). The authors also need to explain their novelty compared to the chiral quasi-BICs proposed for bilayer photonic crystals (10.1103/PhysRevLett.126.073001).

Response: We thank the reviewer for the great suggestion. All the papers listed above are added to the references in the main text. We would like to note that our work is mainly focused on optical vortex generation, which corresponds to the orbital angular momentum of light. In stark contrast, the previous contributions mentioned by the reviewer are focused on different circular polarizations, which correspond to the spin angular momentum of light. Therefore, our work shows distinct physics from previous works.

In order to clarify our novelty, this work is compared with previous pioneering publications in Table R1.

Table R1. Comparison between our work and previous works

	moiré structure	BIC modes	Optical vortex generation
Our work	Yes	Yes	Yes
B. Lou et al.	Yes	No	No
L. Huang et al.	Yes	Yes	No
A. Overvig et al.	No	Yes	No

As shown in Table R1, our paper is the only one working on the optical vortex generated by a moiré structure in TBPC, which has never been discovered before. More specifically, the detailed comparison between this work and the pioneering works mentioned by the reviewer is as follows:

(1) B. Lou et al.

This paper presented the analysis of square lattice TBPC with theoretical prediction of chiral behavior when the twist angle is large. However, their work did not investigate BIC modes hosted in TBPC or vortex generation.

(2) L. Huang et al.

The researchers proposed an effective model of TBPC to construct a moiré quasi-BIC mode, which can greatly enhance the second harmonic generation effect. However, their work did not involve the optical vortex generation.

(3) A. Overvig et al.

By introducing quasi-BIC based on chiral symmetry perturbations, the researchers expanded the range of Fano responses to all possible polarizations, but they did not investigate the moiré system or optical vortex generation.

As a conclusion, our work is different from all these previous works that are focused on the system chirality or quasi-BIC modes. Our novelties can be categorized into three aspects: (1) The optical vortex generation from a moiré photonic system is discovered for the first time; (2) The single-layer BIC mode is found to be the origin of vortex generation in TBPC systems; (3) The moiré structure and its induced interlayer channeling model (“moiré channel”) are found to be the key to this new phenomenon, as the “moiré channel” allows the energy transfer from the BIC mode to the far-field, forming optical vortices. All the above aspects are novel and have never been discovered/discussed in previous works.

Reviewer #2 (Remarks to the Author):

This is a very interesting study. It involves coupling between two honeycomb lattices, where the guided resonance of one honeycomb layer excites the free-space mode, while the BIC mode (with wavevector $K=0$) of the other honeycomb layer is excited by the guide resonance from the former layer. As the $k=0$ mode has an intrinsic topological charge, the radiated free space mode results in optical vortices. This study is the first to discover optical vortices radiating from optical moiré lattices. Rigorous simulation confirmed this finding. I support its publication.

I believe, however, that this work could be presented in a more transparent manner. Please see some of the points mentioned below.

Response: We thank the reviewer for the time in reviewing our paper. We also appreciate the reviewer for pointing out that this work is the first to discover optical vortices radiating from optical moiré lattices. In order to improve the transparency of the work, in the revised manuscript we have added plenty of details with new figures and equations. We hope this revised version provides clearer details to our readers.

Regarding my understanding, the formation of vortices in the far field is rooted in the BIC at $k=0$, as the radiated vortices have the same topological charge as the BIC mode. If this is indeed the case, I suggest that it be explicitly written.

Response: We thank the reviewer for this suggestion. Yes, the radiated vortices have the same topological order as the BIC mode. This is because the far-field radiation is emitted by the guided resonances that are excited by the BIC mode. Hence, the topology of the BIC mode defines the states of polarization in the far-field, which will further decide the topological order of the radiated vortices.

In order to make this conclusion more transparent, we have added the corresponding detailed explanation in the first paragraph in the “Optical vortex generation in TBPC” section in the manuscript, and the current version is as follows:

“Both parts of the light will gain a PB phase depending on the orientation of the SOP of the guided resonance: $\theta_{LCP} = -\theta(\mathbf{k}_1)$ and $\theta_{RCP} = \theta(\mathbf{k}_1)$, where θ_{LCP} and θ_{RCP} denote the phase of the LCP and RCP portion, respectively, and $\theta(\mathbf{k}_1)$ is the angle of the polarization vector. Since BIC is the crossing of nodal lines for x-direction and y-direction portions of the far field radiation, the SOPs will present a vortex around the BIC [34-36]. Thus, the angle of polarization vector $\theta(\mathbf{k}_1)$, as well as θ_{LCP} and θ_{RCP} , will form a vortex structure around the $\mathbf{k}_1 = 0$ point. Hence, the emitted beam will gain the desired spiral phase front, whose topological order is the same as that of the BIC mode and can be described as $l = \mp q$. Here q is the topological order of the BIC, minus and plus signs correspond to LCP and RCP portions, respectively.”

I was initially confused about the difference between BIC and guided resonance. However, I think now I understand that BIC belongs to the guided resonance mode, except that the BIC has a zero wavevector.

Response: We thank the reviewer for this comment. Here, we follow previous work [R6] where BIC is treated differently from a guided resonance. According to their definition, the close relationship between BIC and guided resonances is as follows:

To determine whether a wave can be perfectly confined or not within an open system, they use a simple criterion by examining the wave's frequency: If the frequency falls outside of the continuous spectral range formed by the propagating waves (for example, the frequency lies in the band gap), then the wave can exist as a traditional bound state since there is no pathway for it to radiate away. Conversely, if the wave's frequency falls within the continuous spectrum, there exists a "resonance" that leaks and radiates outwards to infinity. This resonance is called the guided resonance. However, BICs defy this conventional knowledge: It exists within the continuum and coexists with extended waves but remains perfectly confined without any radiation [R6]. As a result, though BIC and the guided resonances both exist in the continuous frequency range, the BIC mode does not radiate energy and thus has an infinite lifetime. On the other hand, the guided resonances radiate energy with a finite lifetime.

In our work, all modes with wave vectors close to the Γ point have frequencies within the continuous spectrum. However, only the mode with zero wavevector (that is, at Γ point) is prohibited from radiation due to its symmetry. Thus, only the mode that has a zero wavevector is a BIC mode, and all the other modes around it are guided resonances.

In response to this comment, we have added detailed explanation in the first paragraph in “Methodology” section of the manuscript as follows:

“Specifically speaking, in a monolayer PhC slab with two-fold rotational symmetry, radiative resonant modes can couple to the far field with frequencies and in-plane wave vectors matching the plane waves in free space. Those radiative resonant modes are usually called guided resonances [38]. Meanwhile, any singlet at Γ point is an exception

since the rotational symmetry of any plane wave with in-plane wave vector $(k_x, k_y) = 0$ mismatches the rotational symmetry of that at Γ singlet [34]. Thus, such a singlet is isolated from the free space, and we call this resonant mode which is decoupled from the free space a symmetry-protected BIC. Even with another identical PhC slab introduced to form a bilayer PhC system, the at- Γ mode will remain a BIC mode, due to the protected two-fold symmetry.”

Reference

[R6] Hsu, Chia Wei, et al. "Bound states in the continuum." *Nature Reviews Materials* 1.9 (2016): 1-13. <https://doi.org/10.1038/natrevmats.2016.48>

With respect to Figure 3, the charge 2 vortices seem already split into two well-separated single-charged vortices? Is this due to the instability of the higher-order vortex at the very beginning (BIC at $K=0$), or this is due to the twisting between layers? Besides, since the twisting angle does not change the birth of the vortices in the far-field, then what happens when the twisting angle takes a much larger one (at the present plots, only small angles are considered).

Response: These are great points. Here our responses are as follows:

(1) The appearance of a charge 2 vortex as two charge 1 vortices is common in the study of optical vortices [R7-R9], whose reason lies in the disturbances in the optical system. Here, we use a hypothetical system to explain it:

In polar coordinates (r, φ) , we set a function $y_1(r, \varphi) = \frac{r}{w} \cdot \exp(-\frac{r^2}{w^2}) \cdot e^{i\varphi}$, where w denotes the waist size of an optical vortex. This is a representative profile of a charge 1 Laguerre-Gaussian beam (we ignore the z related terms and normalization coefficients). As shown in Fig. R4a, this is a charge 1 vortex function with a vortex center at $r = 0$ point.

If there is a small disturbance $\Delta y(r, \varphi) = -\delta \cdot e^{-\frac{r^2}{\sigma^2}}$, the new function will be $y_1'(r, \varphi) = \frac{r}{w} \cdot \exp(-\frac{r^2}{w^2}) \cdot e^{i\varphi} - \delta \cdot e^{-\frac{r^2}{\sigma^2}}$. As we can see in Fig. R4b, its topological charge (q) still equals to 1, because the topological charge of the vortex must be integers and is robust against small disturbance. However, since $y_1'(r, \varphi)$ is non-zero at $r = 0$ point due to the disturbance, the vortex center, which must have zero intensity, is shifted to a new point where $y_1'(r, \varphi) = 0$.

Fig. R4 | Effect of disturbance in vortex structures. (a and b) The normalized electric field and the corresponding phase distribution of a charge 1 vortex without (a) and with disturbance (b). (c and d) The normalized electric field and the corresponding phase distribution of a charge 2 vortex without (c) and with disturbance (d). The black dot denotes the pole of the polar coordinates, and the black arrow denotes the $\theta = 0$ line.

Now we investigate a charge 2 vortex function with a vortex center at $r = 0$ point, which can be described by another function $y_2(r, \varphi) = \left(\frac{r}{w}\right)^2 \cdot \exp\left(-\frac{r^2}{w^2}\right) \cdot e^{i2\varphi}$, as shown in Fig. R4c. After we add a disturbance $\Delta y(r, \varphi) = -\delta \cdot e^{-\frac{r^2}{\sigma^2}}$ to this function, it becomes $y_2'(r, \varphi) = \left(\frac{r}{w}\right)^2 \cdot \exp\left(-\frac{r^2}{w^2}\right) \cdot e^{i2\varphi} - \delta \cdot e^{-\frac{r^2}{\sigma^2}}$. Similar to the charge 1 vortex case, the topological charge remains $q=2$. However, the equation $y_2'(r, \varphi) = 0$ usually have two different solutions in such systems. Thus, the vortex center, whose electric field intensity is zero, splits into two, as shown in Fig. R4d. Such phenomena are commonly observed in simulations and experiments [R7-R9], where high-order optical vortices split into several charge 1 vortices.

(2) To probe the optical vortex generation at larger twists, we further simulate the radiated beam profile from the TBPC systems with larger twist angles, as shown in Fig. R5. We find that, as the twist angle increases, the quality of the emitted optical vortex gets worse. If we further increase the twist angle ($> 8^\circ$), the radiated beam will be chaotic without obvious vortex feature.

Fig. R5 | Optical vortices generation at larger twist angles. The bottom panels are the corresponding phase distributions of the top panels in dashed white circles.

This is because when the twist angle is small, the coupling between the BIC mode in one photonic crystal slab and the guided resonances in the other photonic crystal slab is weak, and thus the BIC mode decays slowly. However, when the twist angle increases, the coupling between the BIC mode and the guided resonances becomes much stronger, leading to a much faster decay rate of the BIC mode as well as a lower quality factor (Q factor), as is depicted in Fig. R6.

Fig. R6 | The Q factor as a function of the twist angle.

In our FDTD simulation, the TBPC system is excited with a pulsed beam. When the Q factor of the BIC mode is high at smaller twist angles, all unwanted guided resonances dissipate much faster than the BIC mode. Thus, after a time period of ~ 5 ps, the radiation by the BIC mode (i.e., the optical vortex emission) will gradually stand out against the chaotic background as all unwanted low-Q modes vanish. On the contrary, when the Q factor of the BIC mode is low at larger twist angles, both the BIC mode and the guided resonances dissipate quickly. Thus, the generated beam will be a mixture of the radiation from both the BIC mode and the guided resonances, resulting in a low-quality optical vortex.

In response to this comment, we have added all the above discussion and numerical results about the optical vortex generation at larger twist angles to the Supplementary Information, section G.

Reference

[R7] Wang, Bo, et al. "Generating optical vortex beams by momentum-space polarization vortices centred at bound states in the continuum." *Nature Photonics* 14.10 (2020): 623-628. <https://doi.org/10.1038/s41566-020-0658-1>

[R8] Carlon Zambon, Nicola, et al. "Optically controlling the emission chirality of microlasers." *Nature Photonics* 13.4 (2019): 283-288. <https://doi.org/10.1038/s41566-019-0380-z>

[R9] Brasselet, Etienne, et al. "Photopolymerized microscopic vortex beam generators: Precise delivery of optical orbital angular momentum." *Applied Physics Letters* 97.21 (2010): 211108. <https://doi.org/10.1063/1.3517519>

Reviewer #3 (Remarks to the Author):

This paper presents a theoretical study of a photonic crystal bilayer, twisted in a manner that mimics twisted bilayer graphene (TBLG). In recent years, TBLG has been of substantial interest in condensed matter physics, because its tunable flat electronic bands can give rise to a wide range of behaviors including superconductivity. The vast majority of TBLG's interesting properties do not carry over into photonics. Superconductivity, for instance, requires an inter-particle pairing mechanism that has no natural analogue in photonics. What the authors propose, here, is that twisted photonic crystals can generate optical vortex beams. This happens because the the twist induces a pair of bound-state-in-the continuum (BIC) slab modes to couple to free space modes in a particular fashion.

Response: We thank the reviewer for the time and review of our paper. We have carefully worked on the helpful comments and suggestions. The point-by-point responses with corresponding changes are as follows:

As far as I know, the radiative mechanism detailed in this paper, involving what the authors call the "Moiré channel", is novel. It is characterized using coupled-mode

theory and verified using full-wave simulations. The theoretical analysis appears to be sound, though too many details are relegated to the supplemental materials, which makes the discussion in the main text difficult to follow. For example, in the discussion around Eqs. (3)-(4), it would be better to provide additional schematics clarifying the definition and role of the SOPs and $\theta(\mathbf{k})$, and to fill in some intermediate steps.

Response: We thank the reviewer for this great suggestion. In the revised manuscript, we have added abundant intermediate steps to the ‘‘Optical vortex generation in TBPC’’ section in the main text with enough details for readers as follows:

‘‘ $|E_{out}\rangle$ is the outgoing wave amplitude, and \mathbf{D} is the coupling coefficient between the resonance and the outgoing plane wave. The orientation of the SOP should be [24]:

$$\begin{pmatrix} d_x \\ d_y \end{pmatrix} = \sqrt{d_x^2 + d_y^2} \cdot \begin{pmatrix} \cos(\theta(\mathbf{k}_1)) \\ \sin(\theta(\mathbf{k}_1)) \end{pmatrix} \quad (4)$$

where $\theta(\mathbf{k}_1)$ is the angle of the polarization vector. By dividing this Jones vector in the basis of left- and right-handed circular polarized (LCP and RCP) unit vectors, the radiated field $|E_{out}\rangle$ of mode \mathbf{k}_1 can be formulated as [39]:

$$|E_{out}\rangle = K\zeta(\mathbf{k}_1)e^{-i\theta(\mathbf{k}_1)}|L\rangle + K\zeta(\mathbf{k}_1)e^{i\theta(\mathbf{k}_1)}|R\rangle \quad (5)$$

„

Also, we have added an additional schematic (Fig. R7) and more explanations to section C in the Supplementary Information to make the derivation of Eqs. (3)-(4) clear.

‘‘For better illustration, we also show a schematic for the vortex generation in Fig. R7. In real space, if we circulate around the beam center, the polarization of the beam will always be close to linear polarization, with the polarization angle changing from 0° to $l \cdot 360^\circ$, where l is the topological order of the BIC. Thus, this trajectory in the real space correspond to the equator of a Poincare sphere in the parameter space, as shown in Fig. R7 [R10]. Since it is on the equator of a Poincare sphere, we can also regard the SOPs as the combination of an LCP portion and an RCP portion with the same amplitude but with different phases. Specifically speaking, $|E_{out}\rangle = e^{-i\theta(\mathbf{k}_1)}|L\rangle + e^{i\theta(\mathbf{k}_1)}|R\rangle$.

Therefore, for a trajectory circulating the beam center in the real space, its SOPs will rotate by $\theta(\mathbf{k}_1)$ while the phases of its LCP portion and RCP portion will gain an extra phase $\mp\theta(\mathbf{k}_1)$. If the trajectory forms a loop in the real space, the phases of its LCP portion and RCP portion will increase by $\mp l \times 2\pi$, forming an optical vortex structure.

Fig. R7 | Schematic for the vortex generation. Different linear states of polarization (double-headed arrows) correspond to different positions on the equator of the Poincaré sphere. S_1 , S_2 and S_3 are the first, second and third Stokes parameters [R11], respectively. The blue (green) double headed arrows correspond to the linear SOPs at the blue (green) dots on the equator of the Poincaré sphere.
”

Reference

[R10] Chipman, Russell A., Wai Sze Tiffany Lam, and Garam Young. *Polarized light and optical systems*. CRC press, 2018.
 [R11] Stokes, George Gabriel. "On the composition and resolution of streams of polarized light from different sources." *Transactions of the Cambridge Philosophical Society* 9 (1851): 399.

There is a notable related prior work. The generation of an optical vortex from a photonic crystal slab was reported in an experimental paper published in Nature Photonics in 2020 [19], by Bo Wang et al. That team also exploited some of the same properties of BICs in photonic crystal slabs, but the precise mechanism was different and not involve Moiré patterns.

Given the existence of at least one other way to generate an optical vortex from a slab geometry, it should be necessary to demonstrate or argue for a clear advantage for the present method. The authors do not appear to make much efforts in this. They do note that the present method can generate vortices of odd order, whereas Ref. [19] is limited to even orders, but given the brevity, it is hard to evaluate how significant this difference

is.

Response: We appreciate this thoughtful comment. As pointed out by the reviewer, the importance of this paper not only lies in technological improvement with better performance, but also lies in the discovery of the novel “moiré channel” mechanism in TBPC system with analytical explanations and in-depth formulation.

In terms of technological improvement compared with previous works (including B. Wang, *Nat. Photonics*, 2020), we have the following advantages:

- (1) The TBPC system presented in our work can be incorporated with micro/nanoelectromechanical systems (MEMS/NEMS) [R12, R13] for *in situ* adjustment of both interlayer separation and twist angle. Such reconfigurability in a vortex generation system leads to the creation of tunable vortex beams with adjustable quality factors, vortex center positions, divergence angles, etc. Furthermore, with the TBPC system, tunable OAM laser with adjustable order numbers could be possible. Such highly reconfigurable optical vortex generators will find vast applications in integrated optical circuits and systems in the future. [R14, R15]
- (2) Different from the monolayer photonic crystal system (B. Wang, *Nat. Photonics*, 2020), our TBPC system is robust against the disturbances of both incident position and incident angle. As shown in Fig. R8, even if we change the incident angle from 0° to 30° , the generated vortex beam remains almost identical, which is emitted from the same position (the AA point) with the same topological order $l = -1$. This is because our optical vortex generation is rooted in the BIC mode which is independent of the incident beam. However, in the monolayer photonic crystal case (B. Wang, *Nat. Photonics*, 2020), their optical vortex generation is based on the properties of the incident beam itself. Thus, their incident angle must be close to 0° . Also, the incident light is even required to be circularly polarized in their work. Compared to their system, ours has much less requirements on the incident light source.

Fig. R8 | Optical vortex generation as a function of incident angles. The bottom panels are the corresponding phase distributions of the top panels in dashed white circles.

(3) As the reviewer pointed out, the TBPC system reported in this work is able to generate optical vortices with odd orders, which is of key importance to many application areas of optical vortices. Specifically speaking, optical vortices supply new degrees of freedom for light and thus are widely utilized to enhance the capacity of data multiplexing. For example, multiple communication channels that originate from the co-existing optical vortices with distinct orders provide more channels for optical communications and quantum information processing [R16, R17]. Compared to the previous work that only generates vortices with even orders, the number of communication channels enabled by our TBPC system is doubled due to the generation of odd-order vortices, which is important and fascinating to the communication industry.

Most importantly, apart from the above technological advantages, our work reports the novel “moiré channel” mechanism for the first time, which reveals a new direction for the study of moiré photonics as well as the correspondence between moiré photonics and moiré van der Waals materials. So far, the majority of moiré photonic crystal papers are trying to reproduce the phenomena which were firstly discovered within the moiré van der Waals systems, such as flat bands and mode localization. However, the vortex generation in TBPC we discovered in this work has no evident corresponding phenomenon in moiré van der Waals systems, which could either be an interesting difference between the two platforms or provide valuable information for future discoveries in twisted van der Waals material systems.

Reference

- [R12] Ho, Chih-Ming, and Yu-Chong Tai. "Micro-electro-mechanical-systems (MEMS) and fluid flows." *Annual review of fluid mechanics* 30.1 (1998): 579-612. <https://doi.org/10.1146/annurev.fluid.30.1.579>
- [R13] Judy, Jack W. "Microelectromechanical systems (MEMS): fabrication, design and applications." *Smart materials and Structures* 10.6 (2001): 1115. <https://doi.org/10.1088/0964-1726/10/6/301>
- [R14] Wei, Bing-yan, et al. "Generating switchable and reconfigurable optical vortices via photopatterning of liquid crystals." *Advanced Materials* 26.10 (2014): 1590-1595. <https://doi.org/10.1002/adma.201305198>
- [R15] Chen, Peng, et al. "Arbitrary and reconfigurable optical vortex generation: a high-efficiency technique using director-varying liquid crystal fork gratings." *Photonics Research* 3.4 (2015): 133-139. <https://doi.org/10.1364/PRJ.3.000133>
- [R16] Barreiro, Julio T., Tzu-Chieh Wei, and Paul G. Kwiat. "Beating the channel capacity limit for linear photonic superdense coding." *Nature physics* 4.4 (2008): 282-286. <https://doi.org/10.1038/nphys919>
- [R17] Bozinovic, Nenad, et al. "Terabit-scale orbital angular momentum mode division multiplexing in fibers." *Science* 340.6140 (2013): 1545-1548. <https://doi.org/10.1126/science.1237861>

The authors also note that the twist is "tunable", but I find this point dubious. In the condensed-matter context, the twist geometry has attracted attention as it provides a relatively simple way to generate new materials (i.e., by making another sample with a different twist angle), bypassing the limitations of chemistry. In photonics, however, there is already substantial freedom in designing and fabricating photonic structures, and numerous ways to exert active control over these structures. Therefore, it is unclear that varying the twist angle is superior to other methods, such as having a slab with 3D patterning.

Response: This is a really good question. Different from the electronics systems, photonic systems offer more flexibility during the designing process due to the advanced capabilities of nanofabrication technologies. However, most of the photonic systems lose their tunability once fabricated since their geometry and material refractive indices are fixed. Here, the term "tunable" refers to the tunability and reconfigurability after fabrication.

For example, micro-opto-electro-mechanical systems (MOEMS) and reconfigurable photonic devices have been a cutting-edge research direction. The large modulation depth and *in situ* tunability of those devices have found a great number of applications in reconfigurable optical systems, including micro-spectrometers [R18], accelerometer [R19], reconfigurable photonic phase arrays [R20], etc.

Compared to the monolayer photonic crystal system, the TBPC system has two more degrees of freedom: the twist angle and interlayer separation. Thus, the TBPC system

serves as a great platform for MOEMS where both the twist angle and the interlayer separation control the properties of generated optical vortex. Once combined with NEMS/MEMS technology, electrical signals could be used to reconfigure TBPC systems for *in situ* modulation of optical vortex generation, which cannot be achieved by monolayer photonic crystal slabs. More details of the realization of such structures are discussed in our answer to the next question.

[R18] El Ahdab, Ranim, et al. "Wide-band silicon photonic MOEMS spectrometer requiring a single photodetector." *Optics Express* 28.21 (2020): 31345-31359.

<https://doi.org/10.1117/12.648909>

[R19] Wu, Yu, et al. "MOEMS accelerometer based on microfiber knot resonator." *IEEE Photonics Technology Letters* 21.20 (2009): 1547-1549.

<https://doi.org/10.1109/LPT.2009.2029556>

[R20] Dong, Kaichen, et al. "A Lithography-Free and Field-Programmable Photonic Metacanvas." *Advanced Materials* 30.5 (2018): 1703878.

<https://doi.org/10.1002/adma.201703878>

If anything, bilayer configurations might be a harder to fabricate, at least in the nanophotonic regime. The design studied by the authors is certainly not one that can be directly implemented in an experiment, since it consists of suspended silicon disks. Alongside the absence of any experimental results, the practicality of the idea is a significant concern.

Response: We thank the reviewer for this important comment! Potential experimental realization approaches are indeed critical to the exploration of this new direction. We have done some further simulations, which confirm that the vortex generation does not require those silicon disks to be suspended in the air (Fig. R9). We can use silicon disks arrays on quartz substrates in the experiment and we are now working on the experimental verification of this work.

Fig. R9 | Optical vortex generation with the existence of quartz substrates. The left panel depicts the normalized amplitude of the LCP portion of the generated vortex, while the right panel is the corresponding phase distribution of the left panel in dashed white circle.

There are 4 approaches that could be exploited in realizing our designed structures.

- (1) Nanofabrication technology. First, silicon thin membranes are transferred to a quartz substrate for further fabrication. Secondly, electron-beam lithography (EBL) and inductively coupled plasma (ICP) etching are utilized to pattern the silicon membrane into honeycomb photonic crystals. Thirdly, plasma enhanced chemical vapor deposition (PECVD) is used to deposit a thin layer of silicon dioxide (SiO_2) onto the photonic crystal. The thickness of PECVD SiO_2 is carefully controlled so that it is half of the desired interlayer separation. Finally, two such devices are clamped together face-to-face with a controlled twist. Solvents with a refractive index similar to quartz may also be applied for a homogenous environment [R21]. The above fabrication process is also schematically illustrated in Fig. R10.

Fig. R10 | Proposed fabrication procedures of TBPC. (a) A silicon thin membrane on a quartz substrate; (b) EBL and ICP patterning of the Si membrane; (c) Deposition of SiO_2 using PECVD; (d) The formation of TBPC without solvents (d) or with solvents (e).

- (2) Nanoscale 3D printing. Now 3D printing technology is quite mature and capable of fabricating photonic crystals, especially for those based on two-photon polymerization lithography [R22]. Our twisted bilayer photonic crystal structure could be optimized for nano 3D-printing. This approach has great potential of low-cost mass production.
- (3) Microwave experiments. Many photonic experiments can be carried out in the microwave region using mesoscale devices. One example is photonic topological insulator, which was demonstrated at 2.6-2.9 GHz using a large copper waveguide [R23]. For a device working at microwave frequencies, it is much easier to meticulously control its geometry because all elements are at the millimeter/centimeter scale.

- (4) Acoustic experiments. The unique correspondence between photonic and acoustic devices has led to many breakthroughs in the acoustic field. For example, after the discovery of photonic topological insulator, acoustic topological insulator was demonstrated using phononic crystals (whose structure resembles that of photonic crystals) [R24]. Similarly, the experimental demonstration of this work can also be found in the acoustic field in the future.

Reference

- [R21] Yang A, et al. 2015. Real-time tunable lasing from plasmonic nanocavity arrays. *Nat. Commun.* 6: 6939. <https://doi.org/10.1038/ncomms7939>
- [R22] Rybin MV, et al. 2015. Band Structure of Photonic Crystals Fabricated by Two-Photon Polymerization. *Crystal.* 5: 61-73. <https://doi.org/10.3390/cryst5010061>
- [R23] Chen WJ, et al. 2014. Experimental realization of photonic topological insulator in a uniaxial metacrystal waveguide. *Nat. Commun.* 5: 5782. <https://doi.org/10.1038/ncomms6782>
- [R24] He C, et al. 2016. Acoustic topological insulator and robust one-way sound transport. *Nat. Phys.* 12: 1124-1129. <https://doi.org/10.1038/nphys3867>

In summary, this is a paper with credible results, but one would expect more work to be done to convince the reader that the design provides a performance advantage over previous designs --- including Ref. [19], but also other possible vortex emitters. Such discussions should be presented in sufficient detail before the paper can be considered for publication in Nature Communications.

Response: We thank the reviewer again for the great comments. In response to these comments, we have added many intermediate steps and new schematics in the main text and Supplementary Information to provide more details for readers. We have also explained the unique advantages of our discovery over previous designs. Most importantly, this work points out a completely new mechanism for optical vortex generation, which is of scientific importance. Meanwhile, we also present several applicable methods to fabricate the TBPC system for experiments. We hope this revised version matches the high standards of Nature Communications.

REVIEWERS' COMMENTS

Reviewer #1 (Remarks to the Author):

In the revised version of the manuscript, the authors have diligently addressed all the concerns and suggestions made by the reviewers. Their responses have been thorough, and they have made necessary modifications accordingly. I am now recommending the publication of the manuscript

Reviewer #2 (Remarks to the Author):

The authors have provided thorough and detailed responses to all of my concerns. In my opinion, the presentation of the work has significantly improved. I fully support its publication in its current form.

Reviewer #3 (Remarks to the Author):

After reading the revised manuscript, as well as the authors' responses to all the referee comments, I am satisfied that the concerns raised in the first review round have been adequately addressed. The paper should be of interest to the photonics research community and is suitable for publication in Nature Communications.

The authors may be able to improve the work by providing more explicit and broader discussion about the usefulness of this design for on-demand on-chip optical vortex generation. At present, the paper is mainly focused on the Moire aspect, and provides scarcely any discussion of how this compares to other vortex generation designs, aside from some passing remarks. This relies on the expertise of the reader to know about the technological significance of the work, which is not ideal. In their response letter, the authors made some good points about the possibility of active control over the vortex generation, and related issues; however, these comments did not seem to have been incorporated into the main text.

Responses to the comments of NCOMMS-23-04686-A

Reviewer #1 (Remarks to the Author):

In the revised version of the manuscript, the authors have diligently addressed all the concerns and suggestions made by the reviewers. Their responses have been thorough, and they have made necessary modifications accordingly. I am now recommending the publication of the manuscript.

Response: We thank the reviewer for the time and efforts in reviewing our manuscript again. It is encouraging that the reviewer believes that our responses have been thorough and recommends the publication of this manuscript.

Reviewer #2 (Remarks to the Author):

The authors have provided thorough and detailed responses to all of my concerns. In my opinion, the presentation of the work has significantly improved. I fully support its publication in its current form.

Response: We thank the reviewer for the time and efforts in reviewing our manuscript again. It is encouraging that the reviewer believes that we have provided thorough and detailed responses to all the concerns.

Reviewer #3 (Remarks to the Author):

After reading the revised manuscript, as well as the authors' responses to all the referee comments, I am satisfied that the concerns raised in the first review round have been adequately addressed. The paper should of interest to the photonics research community and is suitable for publication in Nature Communications.

Response: We thank the reviewer for the time and efforts in reviewing our manuscript again. It is encouraging that the reviewer is satisfied with the responses. We have carefully worked on the helpful suggestions. The point-by-point responses with corresponding changes are as follows:

The authors may be able to improve the work by providing more explicit and broader discussion about the usefulness of this design for on-demand on-chip optical vortex generation. At present, the paper is mainly focused on the Moire aspect, and provides scarcely any discussion of how this compared to other vortex generation designs, aside from some passing remarks. This relies on the expertise of the reader to know about the technological significance of the work, which is not ideal. In their response letter, the authors made some good points about the possibility of active control over the vortex

generation, and related issues; however, these comments did not seem to have been incorporated into the main text.

Response: We thank the reviewer for this great suggestion! In the revised manuscript, we have added abundant content discussing the technological significance of this design to the “Optical vortex generation in TBPC” section in the main text as follows:

“Consequently, no real-space alignment of the position and incident angle is needed in real applications. This is because the optical vortex emission is excited by the self-contained BIC modes rather than the incident light. More direct demonstration of the robustness of optical vortex generation in TBPC can be found in [39]. Thus, in contrast to previous work [24], our system has much less requirements for the incident light source, making TBPC a promising robust platform for stable vortex generation. Moreover, it is possible to incorporate the TBPC system with micro/nanoelectromechanical systems (MEMS/NEMS) for adjustment of both interlayer separation and twist angle. This could lead to the creation of tunable vortex beams with adjustable quality factors, vortex center positions, divergence angles, etc. [39]. Furthermore, with the TBPC system, tunable OAM laser with adjustable order numbers could be possible.”